# Influence of Storage Temperature on Starch Retrogradation and Digestion of Chinese Steamed Bread

**DOI:** 10.3390/foods13040517

**Published:** 2024-02-07

**Authors:** Cheng Li, Shuaibo Shao, Xueer Yi, Senbin Cao, Wenwen Yu, Bin Zhang, Hongsheng Liu, Robert G. Gilbert

**Affiliations:** 1School of Life Sciences, The Chinese University of Hong Kong, Shatin, Hong Kong 999077, China; 2School of Health Science and Engineering, University of Shanghai for Science and Technology, Shanghai 200093, China; 3Department of Food Science & Engineering, Jinan University, Huangpu West Avenue 601, Guangzhou 510632, China; 4School of Food Science and Engineering, South China University of Technology, Guangzhou 510641, China; 5Key Laboratory of Plant Functional Genomics of the Ministry of Education, Jiangsu Key Laboratory of Crop Genetics and Physiology, College of Agriculture, Yangzhou University, Yangzhou 225009, China; 6Queensland Alliance for Agriculture and Food Innovation, Centre for Nutrition and Food Sciences, The University of Queensland, Brisbane, QLD 4072, Australia

**Keywords:** Chinese steamed bread, storage temperature, starch digestion, intermolecular interactions

## Abstract

Chinese steamed bread (CSB), which is widely consumed in East Asia, usually undergoes storage before consumption, but it is unclear how different storage temperatures affect CSB starch retrogradation and digestion properties, which are important for consumers. CSB was stored for 2 days at 25 °C, 4 °C, −18 °C, 4 °C/25 °C temperature cycling (i.e., 24 h at 4 °C, followed by 24 h at 25 °C) and −18 °C/ 25 °C temperature cycling. The results revealed for the first time that more orderly starch double helices are formed when CSB was stored at 4 °C or 4 °C/25 °C. Storage under −18 °C produced lower amounts of, but more heterogenous, starch double helices, with fewer B-type, but more V-type, crystallites. Compared to other storage temperatures, more long-range intermolecular interactions formed between the starch and protein at 4 °C or 4 °C/25 °C. CSB samples showed the slowest starch digestibility when stored at 4 °C. The impact of storage temperature on the starch retrogradation properties and digestibility of CSB also depended on the wheat variety, attributed to differences in the starch molecular structure. These results have significance and practical applications to help the CSB food industry to control starch retrogradation and digestibility. For example, CSB could be stored at 4 °C for 2 days in order to reduce its starch digestibility.

## 1. Introduction

In China and Southeast Asia, steamed bread (CSB) is a traditional staple food [1]. The main ingredients of CSB are flour, yeast, and water. Starch is the main component of CSB, which is largely gelatinized as the bread is cooked [2]. During CSB storage, starch retrogradation will occur, transforming gelatinized starch into relatively ordered structures [3]. This structural transition is important for both the final textural and the nutritional properties of CSB. For example, starch retrogradation can cause staling and reduce product shelf life [4]. On the other hand, retrograded starch is a type-3 resistant starch (RS3), which is preferable for the maintenance of human postprandial glycemic response and a healthy colon [5,6]. The optimization of starch retrogradation properties can thus improve both the textural and nutritional properties of CSB [7].

Storage temperature is a critical factor affecting starch retrogradation properties. Starch retrogradation generally consists of three main steps: nucleation, crystal propagation, and maturation [8]. Nucleation is the rate-limiting step, forming the nuclei for crystal growth and subsequent crystal perfection [9]. Generally, a low storage temperature near the starch glass transition temperature increases the nucleation rate, and operating at a temperature near the starch gelatinization temperature is favorable for crystal growth [10]. Thus, temperature cycling between the starch glass transition and gelatinization temperatures during storage usually enhances the degree and crystal perfection of starch retrogradation [11,12,13]. In addition, storage temperature can also largely determine the retrogradation pattern, as controlled by inter- versus intramolecular interactions; these two types of interactions have different effects on starch digestibility. For instance, gelatinized sago starch forms a much higher amount of intermolecular double helices when stored at −20 °C and 4 °C compared to that stored at room temperature [14]. The intermolecular double helices can form a network gel matrix that physically inhibits the diffusion of starch digestive enzymes and reduces starch digestibility. On the other hand, double helices formed via intramolecular interactions tend to attach to starch digestive enzymes by hydrogen bonding, and thus tend to reduce starch digestibility [15,16].

Although retrogradation and storage are important for CSB properties, the impact of storage temperature on the starch retrogradation and digestion properties of CSB has not been extensively studied. It has been found in baked wheat bread that both nucleation and crystal propagation occur when the bread is stored at 4 °C and 25 °C, whereas only the nucleation step occurs during storage at −18 °C [17]. The storage of bread and cake also increases the slowly digestible and resistant starch contents [18]. However, the results obtained for baked bread and cake cannot be assumed to be applicable to CSB, as the preparation process for CSB is significantly different from those of bread and cake. For instance, CSB is prepared using low-to-intermediate protein-content wheat flour, and by steaming instead of baking (the latter being used for “Western-style” bread). We hypothesize that these different storage temperatures would result in the formation of distinct starch ordered structures in CSB, which would be a major determinant of the starch’s digestibility. A further investigation of the effects of storage temperature on the starch retrogradation and digestion properties of CSB is important for the development of CSB products with both desirable textural and nutritional properties.

This study aims to study the effects of various storage temperatures on intra- versus intermolecular interactions in starch during the cooling/storage of CSB, and their influences on the starch digestibility of CSB. Typical storage temperatures and temperature cyclings will be used, namely room temperature (25 °C), 4 °C, −18 °C, 4 °C/25 °C (meaning 24 h at 4 °C then 24 h at 25 °C), and −18 °C/25 °C. Temperature cycling (e.g., 4 °C/25 °C) is used to enhance the extent of starch retrogradation. The CSB microstructure is recorded via scanning electron microscopy. The crystal morphology of CSB after storage is analyzed via X-ray diffraction. The starch intra- versus intermolecular interactions during CSB storage are characterized through differential scanning calorimetry and small-amplitude oscillatory shear rheological tests. Typically, the melting enthalpy from DSC is determined by the total amount and ordering of both intra- and intermolecular interactions, while storage moduli are controlled by the intermolecular interactions among starch molecules formed during retrogradation [19,20,21,22,23]. These methods have frequently been applied in the literature to analyze the molecular interactions during starch retrogradation [24]. We use the resulting data to analyze the relations between CSB starch retrogradation and digestion. While the study focusses on the effects of storage temperatures on the starch retrogradation and digestion properties of CSB, the findings may have broader implications for the development of other staple foods (e.g., rice) into healthier products with slow starch digestibility, via a better understanding of the general principles in food storage and digestion.

## 2. Materials and Methods

### 2.1. Materials

Three varieties of wheat flour were as used previously [2], and full details are given in that reference. The starch contents for Yongliang4, Xiaoyan6, and Xiaoyan22 were about 71.67%, 70.49%, and 70.62%, respectively. The protein contents for Yongliang4, Xiaoyan6, and Xiaoyan22 were about 13.61%, 13.63%, and 13.25%, respectively. High-activity dry yeast was obtained from Angel Yeast, Xinxiang City, Henan Province, China. The wheat varieties were selected so as to produce CSB with a relatively slow starch digestion rate as well as a soft texture [2].

Pancreatin was obtained from Sigma-Aldrich Chemical Co., Ltd. (St. Louis, MO, USA). The total starch (AA/AMG) assay kit, amyloglucosidase (200 U/mL), and a D-glucose (GOPOD Format) assay kit were obtained from Megazyme International Co., Ltd. (Bray, County Wicklow, Ireland).

### 2.2. Preparation of CSB

To make the CSB, wheat flour (250 g), water (110 g), and high-activity dry yeast (2.5 g) were weighed into a bread mixer (SM500, Ashton Electric Appliance Co., Ningbo, China), and stirred at a constant rate for 10 min in order to develop a dough (Figure 1). This CSB dough was placed in the fermenter (YD-6, Yida Electrical Appliance Co., Ltd., Foshan, China) at 38 °C and 65% humidity for 1 h, then taken out and kneaded for 2 min to discharge gas. Small pieces (~65 g each) were then cut from the dough and made into the shape of a CSB bun. A second fermentation was then continued under the same conditions for 20 min. The CSB dough was finally steamed for 25 min and then allowed to cool for 30 min at room temperature. The cooled CSB was stored at the chosen temperatures (25, 4, −18 °C, 4 °C/25 °C temperature cycling and −18 °C/25 °C temperature cycling) for two days. The notation “4 °C/25 °C” means that, during the first day, the CSB was stored at 4 °C, and subsequently stored at 25 °C for the next day. After storage, CSB was cut into small pieces, frozen at −20 °C overnight, then put into a freeze-dryer (CTFD-10P, Yonghe Chuangxin Electronic Technology Co., Ltd., Qingdao, China) for 24 h. The dried samples were milled and sieved through a 100-mesh sieve (diameter ~0.15 mm) for further use.

### 2.3. Preparation of Wheat Flour Hydrogel

To prepare wheat flour hydrogel, wheat flour (2.5 g) and water (7.5 g) were thoroughly mixed by a glass rod in a glass beaker. The mixture was then placed in an autoclave and gelatinized at a pressure of 121 kPa and a temperature of 105 °C for 30 min. The gelatinized hydrogel was then stored at the above temperatures for 2 days. Wheat flour hydrogel stored at 25 °C for 3 h was used as the control group. These hydrogel samples were analyzed by rheological testing [25].

### 2.4. Scanning Electron Microcopy (SEM)

The CSB samples were placed on the surface of a circular SEM stub using double-sided bonded carbon tape. Subsequently, a gold coating was applied to the samples in a sputter coater for the SEM (Zeiss Gemini2, Hamburg, Germany) analysis with an accelerating voltage of 5 kV.

### 2.5. Differential Scanning Calorimetry (DSC)

CSB powder (4 mg, after retrogradation) and 12 μL of distilled water were weighed into a DSC aluminum pan, which was then equilibrated for a period of 12 h at room temperature. The equilibrated samples were then analyzed with a differential scanning calorimeter (DSC 3, Mettler Toledo, Schwerzenbach, Switzerland). An empty pan was used as the reference. The scanning temperature range was from 20 °C to 100 °C and the heating rate was 10 °C/min. The onset (To), peak (Tp) and conclusion (Tc) temperatures, and the enthalpy (ΔH) were obtained using STARe evaluation software (Version 15.0, Mettler Toledo, Schwerzenbach, Switzerland).

### 2.6. X-ray Diffraction (XRD)

XRD measurements were performed with an X-ray powder diffractometer (D8 Advance, Bruker, Germany) operating at 40 mA and 40 kV. Samples were first equilibrated in a desiccator with constant relative humidity (75%) for 1 week. The diffraction angle 2θ was scanned over the range from 5° to 40°. The step interval was 0.02°. The scanning rate was 3° min^−1^. The relative crystallinity (denoted Xc) was determined through OriginPro2018 64 Bit software (2018b) with the following equation:(1)$$Xc=\frac{\sum_{i=1}^{n}AUCi}{AUCt}$$

Here, AUCt refers to the total area of the diffraction pattern, and AUCi represents the area under the crystalline peak corresponding to index i, both after applying baseline correction.

### 2.7. Rheological Testing

The rheological behavior of wheat flour hydrogel samples was measured by a Discovery HR-1 rheometer (TA instruments, New Castle, DE, USA), following a previous method [25]. A Peltier plate and 20 mm diameter parallel plate were used, with the splint gap set to 2000 μm. Then, excess hydrogel was carefully removed, followed by spreading Vaseline hydrocarbon jelly to the surface edge of the exposed sample in order to restrict moisture evaporation. A strain scan was first conducted from a strain percentage of from 0.1% to 4% at 25 °C and constant frequency of 1 Hz to determine the linear viscoelastic region. A temperature scan was then performed from 30 °C to 95 °C at a strain value of 4% (within the linear viscoelastic range), heating rate of 5 °C/min, and frequency of 1 Hz. Storage modulus (G′), loss modulus (G″), and tan δ (G″/G′) were recorded.

In addition, the following quantities were measured: the G′ value was 30 °C (G′30), the G′ value was 95 °C (G′95), and the change in the G′ value ranged from 30 °C to 95 °C (ΔG′(30–95)). G′30 is a measure of the overall number and ordering of the intermolecular interactions caused by starch and protein molecules during retrogradation, G′95 is a measure of the number and ordering of the long-range intermolecular interactions, and ΔG′(30–95) a measure of the short-range intermolecular interactions between starch and protein molecules [26,27,28].

### 2.8. In Vitro Starch Digestion Assay

The in vitro digestion testing of retrograded CSB samples was performed using a published method [2,15]. In brief, the CSB starch content was measured with the total starch (AA/AMG) assay kit, following the protocol for samples that include resistant starch but not D-glucose and maltodextrin. Then, a CSB sample (50 mg) was transferred into a centrifuge tube (50 mL) and mixed with distilled water (2 mL). Enzyme solution (8 mL) in sodium acetate buffer solution (0.2 M, pH 6.0) containing 16.7 μL of amyloglucosidase and 0.33 mg of pancreatin was then transferred to the sample mixture. The sample mixture was then incubated in a water bath at 37 °C, and 0.1 mL aliquots were taken out at different time intervals (0, 5, 10, 15, 20, 30, 45, 60, 90, 120, and 180 min) into a 2 mL centrifuge tube with 0.9 mL of absolute ethanol to stop the digestion. The mixture was then centrifuged at 1000× *g* for 10 min, and glucose concentration from the supernatant was determined with the GOPOD method (Megazyme), and then converted into the amount of digested starch.

### 2.9. Fitting to First-Order Kinetics

Starch digestograms frequently adhere to the integrated form of the first-order loss equation [29]:(2)$$C\left(t\right)=C_{0}+\left(C_{\infty }-C_{0}\right)\left(1-e^{-kt}\right)$$

Here, *C*(*t*) is the fraction of starch digested at time *t*. *C*_∞_ is the fraction of starch digested at infinite reaction time, *C*_0_ is the fraction of starch digested at time 0, and *k* is the starch digestion rate coefficient. To see if the present system indeed follows first-order kinetics, a logarithm of slopes (LOS) test [30,31] was carried out (see below). The values of *k* and *C*_∞_ were finally calculated via non-linear least-squares fitting [32] to digestograms.

In order to determine whether multiple starch digestible components are present in CSB samples, an LOS plot was performed, using Equation (3), as follows:(3)$$ln\frac{dC_{t}}{dt}=ln[k(C_{\infty }-C_{0})]-kt$$

This used numerical derivatives from the experiment data [30]. Observing multiple linear regions in such a plot indicates the presence of multiple components.

### 2.10. Statistical Analysis

The means and standard deviations were determined in Excel 2019 for all parameters. A statistical analysis was implemented using SPSS software (version 16) to assess the significant differences between the parameters through a one-way ANOVA with a Duncan test (*p* < 0.05).

## 3. Results and Discussion

### 3.1. Microstructure of CSB

Typical examples of the microstructure of CSB after retrogradation are shown in Figure 2, illustrating the effects of different storage temperatures on the CSB microstructures. Objects appearing to be starch granules were seen for some wheat CSB samples, designated by their size and shape into type A (with a disc-like shape and diameter of about 6–50 µm) and type B (with a spherical shape and diameter of about 1–6 µm) [33]. However, these are probably not actually granules but granule ghosts (i.e., the shells of the granules), as it has been shown that, after steaming, there are no endothermic peaks in the DSC measurements of CSB powders [2]. CSB samples stored at 4 °C showed relatively smaller and more scattered particles compared to those stored at other temperatures. However, there were no visual differences with CSB samples stored at other temperatures, or among the three varieties stored at a given temperature. This indicates that the microstructure is not the dominant factor determining the digestion characteristics of CSB samples, as discussed in following sections.

### 3.2. Retrogradation Properties Analyzed by DSC

Table 1 shows the DSC parameters for raw wheat flours and retrograded CSB samples. These parameters are related to both intra- and intermolecular interactions among the starch molecules formed during retrogradation [19]. Generally, the value of Tc—To reflects the homogeneity of starch double helices/ entanglements, where less stable double helices melt at a lower temperature (To) while remaining more stable double helices melt at a higher temperature (Tc) [34]. The melting enthalpy ΔH is the energy required to disrupt ordered structures within starch granules, and reflects the overall ordering and amount of starch double helices or entanglements in these retrograded CSB samples [34].

Consistent with prior research findings [15], the retrograded CSB samples showed a much lower melting temperature and enthalpy compared to raw wheat flours. It could be ascribed to the crystallites developed during retrogradation were relatively less stable compared to those formed in the raw starch granules. CSB samples stored at 4 °C or 4 °C/25 °C had higher melting enthalpies but lower melting temperature ranges than other samples, suggesting that more orderly crystallites were formed under these conditions [23]. This is consistent with the literature [11,12,13], in which it was shown that a storage temperature cycling between 4 °C and room temperature can enhance the nucleation and crystal propagation rate during retrogradation. On the other hand, although CSB samples stored at −18 °C had the lowest melting enthalpy, they showed the highest melting temperatures and melting temperature range. This indicates that although a smaller number of crystallites was formed at −18 °C, these crystallites were more thermally stable.

The retrogradation behavior of CSB varied with the variety of the wheat flour, as seen in the distinct DSC parameters of CSB presented in Table 1. For example, Xiaoyan6 wheat flour showed a CSB with a higher melting enthalpy compared to Xiaoyan22 wheat flour. This is possibly due to the longer amylopectin external chains of Xiaoyan6 [2], which could lead to a faster retrogradation rate [2]. This highlights the importance of considering wheat flour composition and structure when studying CSB retrogradation properties; further research is needed to yield insights into the mechanisms underlying this process.

### 3.3. Crystal Structure Analysis

XRD data and crystallinity parameters are summarized in Figure 3 and Table 2, which show the polymorph (A-, B- or V-type crystallinity) and degree of starch crystallinity. As expected, raw wheat flours all exhibited an A-type crystallinity diffraction pattern, with characteristic peaks around 2θ of 15°, 17°, 18°, 23°, and 26° [35]. A minor peak at 2θ~20° was also seen in raw wheat flour, which is related to the V-type crystallinity [17,35,36]. On the other side, B-type crystallinity appeared after the retrogradation of the CSB samples, with peaks with 2θ~7.5°, 13°, 15°, 17°, 22.5°, and 24°, which is consistent with the literature [37,38,39]. A significantly higher peak at 2θ of 20° was observed for retrograded CSB samples compared to the raw wheat flour, indicating that more amylose–lipid complexes were developed during the retrogradation process [40].

Retrograded CSB samples all showed a lower total crystallinity compared to raw wheat flours (Table 2), suggesting that the retrograded starch structure during retrogradation was less stable than that from raw wheat flours. This is in line with the DSC results. Although the crystallinity of the retrograded CSB samples varied with storage temperature and wheat variety, CSB samples stored at −18 °C showed the lowest total and B-type crystallinities and the highest V-type crystallinity compared to the other samples. This shows that, in CSB, although storage at −18 °C inhibits the overall starch retrogradation process, it promotes the development of amylose–lipid complexes.

Similarly, the crystallinity degree of retrograded CSB is influenced by the chemical composition and structural differences of macro-/micronutrients in wheat flours, as evidenced by the different fractions of crystallinity in CSB prepared from various wheat flours (Table 2). For instance, Yongliang4 CSB generally exhibited a higher degree of crystallinity (especially with V-type crystallinity) compared to other CSB samples, possibly due to differences in their starch molecular structures. Yongliang4 has relatively longer amylopectin external chains [2], which are positively correlated with a faster retrogradation rate [2].

### 3.4. Rheological Properties of Flour Hydrogel

The rheological test results for the retrograded wheat flour gels are summarized in Figure 4. For pure starch hydrogels, storage moduli (G′) are positively related to the quantity and ordering of intermolecular interactions, whereas elastic moduli (G″) are controlled by the energy consumption from the friction and movement of starch molecules during the rheological test [20,21,22,23]. Thus, a drop (ΔG′) in the storage modulus from a temperature sweep test (e.g., from 30 °C to 95 °C) is probably due to the short-range starch intermolecular interactions formed during retrogradation, while the values of the storage modulus at 30 °C (G′30) and 95 °C (G′95) reflect the overall starch intermolecular interactions and long-range intermolecular interactions that are not disrupted during the temperature sweep test, respectively [11]. However, compared to pure starch, the retrogradation behavior of wheat flour is much more complex, as it involves not only interactions among starch molecules but also interactions between starch and protein, via hydrogen bonds and hydrophobic forces [40,41,42]. Therefore, G′30, G′95, and ΔG′(30–95) were taken in this study to reflect, respectively, the overall intermolecular interactions and long-range and short-range intermolecular interactions between starch and protein molecules (Table 3).

For all wheat flour gels, G′ was much higher than G″ (i.e., tan δ < 1), indicating these gels showed an elastic behavior rather than liquid-like behavior (Figure 4) [23]. This could be because of the high number of intermolecular interactions developed among starch and protein molecules during retrogradation. Consistent with the literature, all wheat flour gels showed a downward trend in both G′ and G″ during the rheological temperature sweep test, ascribed to the disruption of short-range intermolecular interactions between starch and protein molecules [14,41].

The rheological parameters showed a wide range of values for gels stored at different temperatures (Table 3). Generally, the G′30 and ΔG′(30–95) values of samples were higher when stored at −18 °C and −18 °C/25 °C compared to those samples stored at other temperatures. This indicates that more short-range intermolecular interactions were formed at these temperatures. The G′95 values of samples were generally higher when stored at 4 °C or 4 °C/25 °C than the others, suggesting that more long-range intermolecular interactions were formed under these temperatures. One possible reason for this is that starch and protein molecules are more flexible at 4 °C than at −18 °C, allowing for them to form long-range interactions. In addition, different wheat varieties showed distinct rheological properties, probably related to their different starch molecular structures [2]. For example, Xiaoyan6 had a much higher G′30 value than other wheat varieties, possibly because it has longer amylopectin trans-lamellar chains, which can engage in the development of intermolecular interactions during retrogradation [2].

### 3.5. In Vitro Digestion Characteristics of CSB

The digestograms and corresponding fitting parameters for different samples are shown in Figure 5. LOS plots were used to determine the number of starch digestible components (Appendix A) [38], and suggested that there was only a single component in the starch digestograms for these retrograded CSB samples. Simple first-order kinetics was applied to obtain kinetic parameters [9]. The division of starch into RDS, SDS, and RS was not applied in this study, as it would provide redundant information because the whole digestogram is fitted by first-order kinetics with a single rate coefficient.

As expected, retrograded CSB showed a significantly slower digestion rate and final extent than freshly prepared CSB (Figure 5). The maximum starch digestion percentage for the freshly prepared CSB sample reached about 95%, probably because the loss of orderly starch structures during steaming [2]. However, the maximum starch digestion percentage was decreased by at least 5% by the retrogradation treatment. Among the different storage temperatures, CSB samples stored at 4 °C showed the slowest digestion rate and smallest maximum digestion extent compared to samples stored at other temperatures. On the other hand, the CSB sample stored at −18 °C exhibited the fastest digestion rate compared to other samples. CSB samples stored at 4 °C had relatively high melting enthalpies, low melting temperature ranges, and high G′95 values, while CSB samples stored at −18 °C had a relatively low melting enthalpy, high melting temperature range, high V-type crystallinity, low B-type crystallinity, and high values of G′30 and ΔG′(30–95); all these factors indicate that, in retrograded CSB samples, (1) more orderly starch double helices/entanglements and long-range intermolecular interactions by starch and protein molecules will result in slower starch digestion rates, while (2) V-type crystallinity and short-range intermolecular interactions by starch and protein molecules would not be significant factors in slowing down starch digestion rate.

Wheat variety also affects the starch digestibility in retrograded CSB samples because of molecular structural differences; it is important to remember that genetics + environment together control the starch structure, and the structure controls properties such as those considered here, while variety (genetics) and environment do not directly control these properties. Xiaoyan22 had the smallest maximum starch digestion extent compared to the other two wheat varieties. This indicates that the storage temperature has different effects on CSB than various other wheat varieties, possibly because of their molecular structural differences in starch (e.g., Xiaoyan22 had the highest amylose content among the three wheat varieties) [2].

## 4. Conclusions

The effects of storage temperature on starch retrogradation and digestibility in CSB were explored in this study. The results showed for the first time that CSB samples had a higher melting enthalpy and lower melting temperature range than other samples when stored at 4 °C or 4 °C/25 °C. On the other hand, CSB samples had the lowest melting enthalpy, the highest melting temperatures and melting temperature range, and the lowest total and B-type crystallinity, but the highest V-type crystallinity compared to other samples when stored at −18 °C. Wheat flour gels stored at −18 °C and −18 °C/25 °C had relatively higher G′30 and ΔG′(30–95) values, while those stored at 4 °C or 4 °C/25 °C generally had higher G′95 values than other samples. Finally, CSB samples after storage at 4 °C showed the slowest starch digestibility. All these trends are ascribed to structural differences. For example, Xiaoyan22 had the smallest maximum starch digestion extent compared to the other two wheat varieties, probably due to its having the highest amylose content.

These results can assist the food industry in implementing different storage regimes to effectively regulate the starch retrogradation properties and digestibility in CSB. For example, storage at 4 °C could be applied in the food industry to reduce the starch digestibility in CSB. This could be applied at a plant scale in the future to prove its practical significance.

Finally, there are many aspects that could be explored further based on the findings from this study. For example, texture is another important attribute of CSB, determining its acceptability by consumers. It is not known if CSB with both slow starch digestibility (which is desirable from a health perspective) and a desirable texture could be produced via different retrogradation regimes. Furthermore, if starch is digested slowly enough to reach the colon (i.e., RS), it could affect human health via interacting with gut microbiota. Therefore, it merits further investigation on if, and how, CSB prepared by different retrogradation conditions can affect human gut microbiota.

## Figures and Tables

**Figure 1 foods-13-00517-f001:**
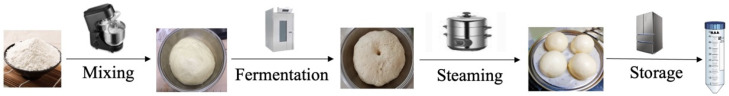
Procedure for the preparation of CSB for the in vitro starch digestion assay: (1) the mixing of wheat flour, water, and yeast; (2) fermentation of CSB dough; (3) steaming; and (4) storage at different temperatures.

**Figure 2 foods-13-00517-f002:**
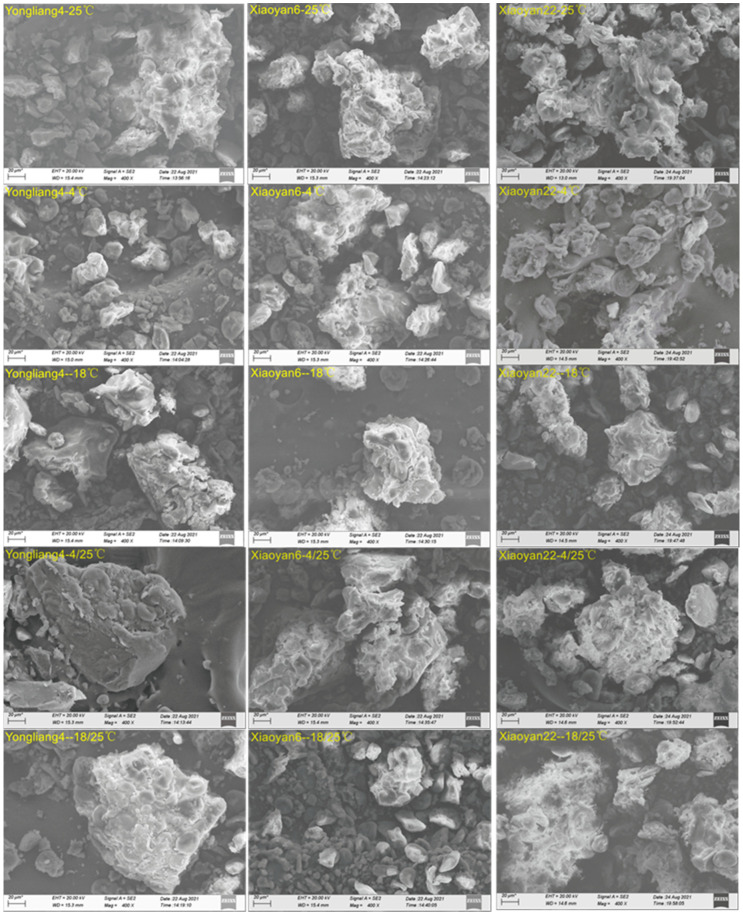
SEM micrographs of CSB samples stored at 25 °C, 4 °C, −18 °C, 4 °C/25 °C, and −18 °C/25 °C. The scale bar is 20 μm.

**Figure 3 foods-13-00517-f003:**
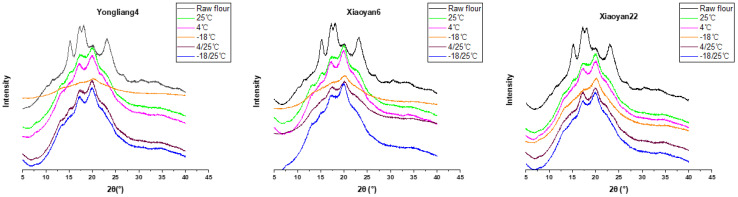
XRD curves of CSB samples at different storage temperatures.

**Figure 4 foods-13-00517-f004:**
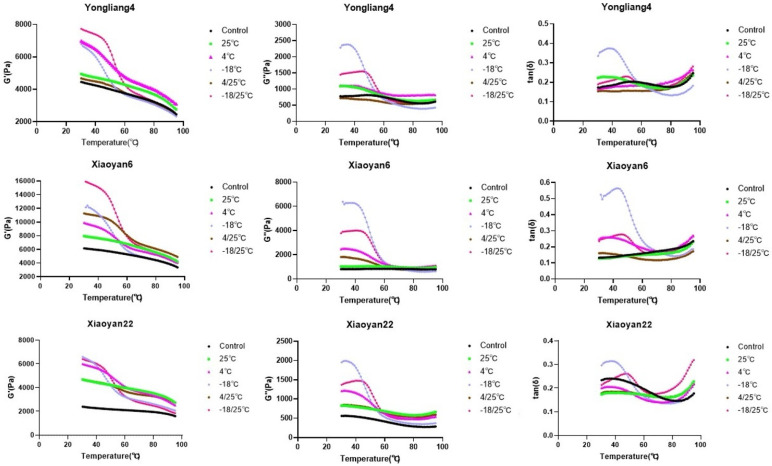
Changes in G′, G″ and tan δ from the rheological temperature sweep test for wheat flour gels stored at different temperatures.

**Figure 5 foods-13-00517-f005:**
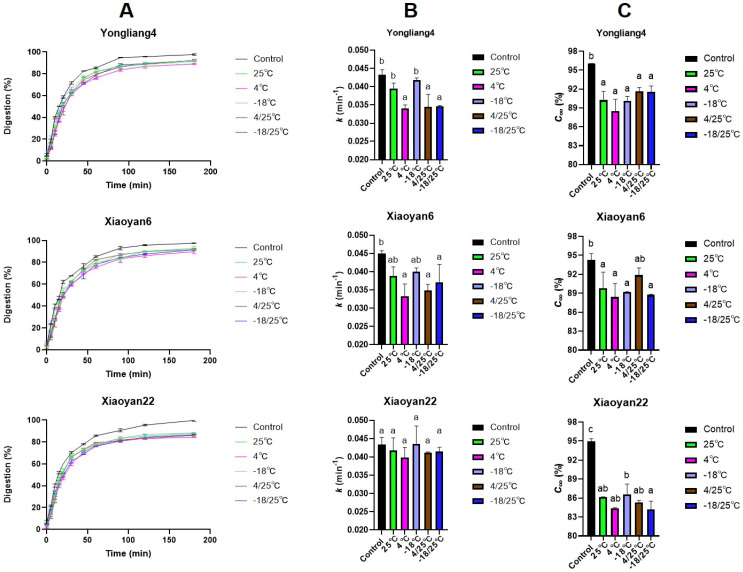
Digestion curves (**A**) and fitting parameters (**B**,**C**) for retrograded CSB samples stored under different temperatures. Different letters above bars in graphs B and C indicate significant differences at the level of *p* < 0.05.

**Table 1 foods-13-00517-t001:** Thermal data for CSB samples used here.

Flour Variety	Temperature	To (°C)	Tp (°C)	Tc (°C)	Tc—To (°C)	ΔH (J/g)
Yongliang4	Raw flour	56 ± 1c	62 ± 1c	68 ± 2c	12 ± 3a	7 + 2b
	25 °C	42 ± 1a	49 ± 0ab	58 ± 1ab	16 ± 2ab	2 + 1a
	4 °C	42 ± 1a	49 ± 0ab	55 ± 0a	13 ± 1a	2 + 0a
	−18 °C	45 ± 2 ab	52 ± 1ab	65 ± 1bc	20 ± 1b	1 + 0a
	4 °C/25 °C	41 ± 1a	47 ± 1a	52 ± 3a	11 ± 5a	1 + 1a
	−18 °C/25 °C	50 ± 6bc	54 ± 5b	61 ± 8abc	11 ± 2a	1 + 0a
Xiaoyan6	Raw flour	58 ± 0b	63 ± 0b	68 ± 0c	10 ± 0a	6 + 0c
	25 °C	43 ± 2 a	51 ± 0a	60 ± 4b	17 ± 6bc	2 + 0b
	4 °C	45 ± 0a	50 ± 0a	55 ± 1a	10 ± 0ab	1 + 0ab
	−18 °C	45 ± 2a	57 ± 8ab	67 ± 0c	22 ± 2c	1 + 0a
	4 °C/25 °C	45 ± 1a	51 ± 0a	56 ± 1a	11 ± 0ab	2 + 1b
	−18 °C/25 °C	46 ± 0a	52 ± 0a	57 ± 0ab	11 ± 0ab	1 + 0ab
Xiaoyan22	Raw flour	56 + 2c	62 + 0b	67 + 2b	12 + 4a	5 + 0d
	25 °C	46 + 0b	52 + 0a	57 + 0a	11 + 0a	1 + 0b
	4 °C	45 + 1ab	51 + 0a	55 + 0a	10 + 1a	1 + 0b
	−18 °C	45 + 0b	52 + 2a	69 + 2b	23 + 2b	0 + 0a
	4 °C/25 °C	43 + 1a	50 + 1a	58 + 0a	15 + 1a	2 + 0c
	−18 °C/25 °C	47 + 0b	52 + 0a	57 + 0a	11 + 0a	1 + 0b

Note: values are shown as mean ± SD, with different letters indicating a significant difference at the level of *p* < 0.05.

**Table 2 foods-13-00517-t002:** XRD characteristics of retrograded CSB samples at different storage temperatures.

Flour Variety	Temperature	Crystallinity Polymorph	V-Type Crystallinity (%)	B-Type Crystallinity (%)	Total Crystallinity (%)
Yongliang4	Raw flour	A + V	1.20		11.30
	25 °C	B + V	2.58	3.07	5.65
	4 °C	B + V	2.85	2.39	5.24
	−18 °C	B + V	3.62	1.36	4.98
	4 °C/25 °C	B + V	2.99	3.13	6.12
	−18 °C/25 °C	B + V	3.12	2.94	6.06
Xiaoyan6	Raw flour	A + V	0.96		12.12
	25 °C	B + V	2.24	3.34	5.58
	4 °C	B + V	2.21	3.07	5.28
	−18 °C	B + V	3.98	0.70	4.68
	4 °C/25 °C	B + V	2.14	2.46	4.60
	−18 °C/25 °C	B + V	2.88	2.19	5.07
Xiaoyan22	Raw flour	A + V	0.67		11.80
	25 °C	B + V	2.60	2.98	5.58
	4 °C	B + V	2.41	2.64	5.06
	−18 °C	B + V	2.75	1.12	3.87
	4 °C/25 °C	B + V	2.29	3.55	5.84
	−18 °C/25 °C	B + V	2.39	3.13	5.52

**Table 3 foods-13-00517-t003:** Rheological parameters for wheat flour gels stored at different temperatures.

Flour Variety	Temperature	ΔG′(30–95)	G′30	G′95
Yongliang4	Control	2086 + 213a	4459 + 390a	2373 + 178a
	25 °C	2201 + 601a	4963 + 772a	2762 + 186b
	4 °C	3849 + 1327b	6952 + 144b3	3103 + 115c
	−18 °C	4272 + 106bc	6599 + 67b	2327 + 116a
	4 °C/25 °C	2259 + 290a	4703 + 434a	2444 + 145a
	−18 °C/25 °C	5283 + 1147c	7706 + 1340b	2423 + 193a
Xiaoyan6	Control	2913 + 213a	6153 + 350a	3240 + 138a
	25 °C	3765 + 283a	7959 + 283ab	4194 + 256c
	4 °C	5701 + 398b	9734 + 660bc	4034 + 262bc
	−18 °C	7448 + 875c	10,832 + 1020c	3384 + 145ab
	4 °C/25 °C	6399 + 1941bc	11,291 + 2599c	4892 + 658d
	−18 °C/25 °C	11,547 + 794d	15,656 + 797d	4109 + 484c
Xiaoyan22	Control	841 + 138a	2390 + 243a	1549 + 106a
	25 °C	1978 + 435b	4711 + 542b	2733 + 173c
	4 °C	3511 + 585c	6023 + 1052c	2512 + 474c
	−18 °C	4427 + 244d	6489 + 248c	2062 + 5b
	4 °C/25 °C	1972 + 111b	4701 + 123b	2730 + 21c
	−18 °C/25 °C	4538 + 365d	6375 + 468c	1837 + 157ab

Note: The values are presented as mean ± SD, with different letters in same columns are significantly different at the level of *p* < 0.05.

## Data Availability

The raw data supporting the conclusions of this article will be made available by the authors on request.

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
