# Peer review of "Influence of Storage Temperature on Starch Retrogradation and Digestion of Chinese Steamed Bread"

_foods, 2024, doi:10.3390/foods13040517_

Round 1

Reviewer 1 Report

Comments and Suggestions for Authors

Novelty is not high enough to even evaluated. Many experiemnts to deep understand such effects are missing. 

Reviewer 2 Report

Comments and Suggestions for Authors

With the present work, the authors intend to investigate the influence of storage temperatures on the retrogradation of starch and its potential exploitation for the food industry.

Paper topic is in line with the broader scope of the journal: however, paper structure and conceptualization, especially in its introduction, appears rather technical-theoretical when considering the methodologies used and the variables observed as results.
Therefore, for a greater understanding of the experimental results subsequently exposed in the paper, it would be useful to restructure the introductory part (starting from line 67) by explaining with more detail, also with support of new references, the practical meaning of each analytical technique used in the paper.

As for the paper scientific content, the general impression is of a complex methodological approach.

Presentation of results look somehow confused, and rheological graphs (figure 2 has to be broken at least into two parts to achieve proper dimensions) look unreadable in the present form.

Conclusions are very sparse, and dismissing the opportunities for improvement for the food industry in a single sentence appears to be too little for journal focus.

In conclusion, the paper presents a considerable amount of experimental data to support the research objective, however a significant effort is necessary on the part of the authors to increase its readability, currently limited by the excessive technicality in data presentation, discussion and conclusions.

Comments on the Quality of English Language

The English form would benefit from a native speaking review.

Reviewer 3 Report

Comments and Suggestions for Authors

The manuscript, titled "Effects of Storage Temperatures on Starch Retrogradation and Digestion of Chinese Steamed Bread," explores the impact of different storage temperatures on the starch retrogradation and digestion properties of Chinese steamed bread (CSB). Additionally, the manuscript highlights variations in starch retrogradation and digestibility, influenced by the wheat flour variety used. These results hold significance for the food industry, providing practical applications to control starch retrogradation and digestibility in CSB. Despite the manuscript's well-prepared nature, it is important to note the presence of minor shortcomings.

General remarks: As a suggestion for improvement, correct the in-text reference style to adhere to the publisher's guidelines.

I would recommend refining lines 67-77 by focusing on articulating the hypothesis rather than delving into the methods.

In section 2.1., I suggest expanding the information regarding the wheat flour used.

Section 2.2 on Chinese steamed bread preparation and storage cycling, I recommend enhancing the clarity of presentation, preferably through a graphical scheme. This visual aid can significantly improve readability and help readers grasp the process more effectively.

At line 101, provide clarification on the sieve used.

I recommend a comprehensive revision of the entire method section, focusing on incorporating essential details about the apparatus and reagents used, including manufacturer data.

The discussion pertaining to the bread microstructure appears incomplete and would benefit from additional supplementation. Please consider providing a more thorough analysis in this section to ensure a comprehensive exploration of this crucial aspect of your research.

Figure 2 is challenging to decipher, and I recommend either dividing the information for better readability, presenting enlarged details, or relocating the data to the supplementary material.

The data presented in Figure S1 of the supplementary material is both significant and effectively illustrates the obtained results. I recommend considering the transfer of this valuable information from the supplementary material to the main text.

Table 3 is less legible, and I am concerned about the high standard deviations of the presented results. I recommend a thorough reevaluation of these values and consider presenting the results in a more readable format.

In the conclusions section, it is advisable to provide a more comprehensive summary of the entire manuscript. Additionally, consider incorporating information about recommendations based on the obtained results and outlining perspectives for future research stemming from these findings.

Comments on the Quality of English Language

Minor editing of English language required

Reviewer 4 Report

Comments and Suggestions for Authors

The article studied the effects of storage temperatures on starch retrogradation and digestion of steamed Chinese bread. I believe that this information will not have a great global impact. This is because the product is mainly consumed in East Asia.

Nutritional, texture, sensory, flavor and microbiological aspects were not considered in this study. I suggest approaching these parameters to correlate them with the results obtained in this study (storage temperatures on starch retrogradation and digestion).

The article does not focus its study on new or novel methods that can be easily applied in the food industry to control starch retrogradation. At the same time, this was a study carried out on a laboratory scale that would need more research to confirm its significance and practical application.

The size of the starch granules should be included in the discussion.

Rapidly digestible starch (RDS), slowly digestible starch (SDS), total digestible starch (TDS) and resistant starch (RS) should be included in the discussion. 

Comments on the Quality of English Language

Minor editing of English language required

Round 2

Reviewer 1 Report

Comments and Suggestions for Authors

It could be accepted in its current form.

Reviewer 2 Report

Comments and Suggestions for Authors

The authors have addressed most of the reviewer's comments. Furthermore, the text of the entire paper has been revised. It is therefore concluded that in its current form the paper content may be suitable for publication.

Reviewer 4 Report

Comments and Suggestions for Authors

no comment

Comments on the Quality of English Language

no comment